# GENERATIVE MULTI-FLOW NETWORKS: CENTRALIZED, INDEPENDENT AND CONSERVATION

## ABSTRACT

Generative flow networks utilize the flow matching loss to learn a stochastic policy for generating objects from a sequence of actions, such that the probability of generating a pattern can be proportional to the corresponding given reward. However, existing works can only handle single flow model tasks and cannot directly generalize to multi-agent flow networks due to limitations such as flow estimation complexity and independent sampling. In this paper, we propose the framework of generative multi-flow networks (GMFlowNets) that can be applied to multiple agents to generate objects collaboratively through a series of joint actions. Then, the centralized flow network algorithm is proposed for centralized training GMFlowNets, while the independent flow network algorithm is proposed to achieve decentralized execution of GMFlowNets. Based on the independent global conservation condition, the flow conservation network algorithm is then proposed to realize centralized training with decentralized execution paradigm. Theoretical analysis proves that using the multi-flow matching loss function can train a unique Markovian flow, and the flow conservation network can ensure independent policies can generate samples with probability proportional to the reward function. Experimental results demonstrate the performance superiority of the proposed algorithms compared to reinforcement learning and MCMC-based methods.

## 1 INTRODUCTION

Generative flow networks (GFlowNets) Bengio et al. (2021b) can sample a diverse set of candidates in an active learning setting, where the training objective is to approximate sample them proportionally to a given reward function. Compared to reinforcement learning (RL), where the learned policy is more inclined to sample action sequences with higher rewards, GFlowNets can perform better on exploration tasks. Since the goal of GFlowNets is not to generate a single highest-reward action sequence, but to sample a sequence of actions from the leading modes of the reward function Bengio et al. (2021a). Unfortunately, currently GFlowNets cannot support multi-agent systems.

A multi-agent system is a set of autonomous, interacting entities that share a typical environment, perceive through sensors and act in conjunction with actuators Busoniu et al. (2008). Multi-agent reinforcement learning (MARL), especially cooperative MARL, are widely used in robotics teams, distributed control, resource management, data mining, etc Zhang et al. (2021); Canese et al. (2021); Feriani & Hossain (2021). Two major challenges for cooperative MARL are scalability and partial observability Yang et al. (2019); Spaan (2012). Since the joint state-action space grows exponentially with the number of agents, coupled with the environment's partial observability and communication constraints, each agent needs to make individual decisions based on local action observation history with guaranteed performance Sunehag et al. (2017); Wang et al. (2020); Rashid et al. (2018). In MARL, to address these challenges, a popular centralized training with decentralized execution (CTDE) Oliehoek et al. (2008); Oliehoek & Amato (2016) paradigm is proposed, in which the agent's policy is trained in a centralized manner by accessing global information and executed in a decentralized manner based only on local history. However, extending these techniques to GFlowNets is not straightforward, especially in constructing CTDE-architecture flow networks and finding IGM conditions for flow networks worth investigating.

In this paper, we propose Generative Multi-Flow Networks (GMFlowNets) framework for cooperative decision-making tasks, which can generate more diverse patterns through sequential joint ac-

tions with probabilities proportional to the reward function. Unlike vanilla GFlowNets, our method analyzes the interaction of multiple agent actions and shows how to sample actions from multi-flow functions. We propose the Centralized Flow Networks (CFN), Independent Flow Networks (IFN) and Flow Conservation Networks (FCN) algorithms based on the flow matching condition to solve GMFlowNets. CFN regards multi-agent dynamics as a whole for policy optimization, regardless of combinatorial complexity and the demand for independent execution, while IFN suffers from the flow non-stationary problem. In contrast, FCN takes full advantage of CFN and IFN, which is trained based on the independent global conservation (IGC) condition. Since FCN has the CTDE paradigm, it can reduce the complexity of flow estimation and support decentralized execution, which is beneficial to solving practical cooperative decision-making problems.

**Main Contributions:** 1) We are the first to propose the concept of generative multi-flow networks for cooperative decision-making tasks; 2) We propose three algorithms, CFN, IFN, and FCN, for training GMFlowNets, which are respectively based on centralized training, independent execution, and CTDE paradigm; 3) We propose the IGC condition and then prove that the joint state-action flow function can be decomposed into the product form of multiple independent flows, and that a unique Markovian flow can be trained based on the flow matching condition; 4) We conduct experiments based on cooperative control tasks to demonstrate that the proposed algorithms can outperform current cooperative MARL algorithms, especially in terms of exploration capabilities.

## 2 GMFLOWNETS: PROBLEM FORMULATION

### 2.1 PRELIMINARY

Let $F : \mathcal{T} \mapsto \mathbb{R}^+$ be a trajectory flow Bengio et al. (2021b), such that $F(\tau)$ can be interpreted as the probability mass associated with trajectory $\tau$. Then, we have the corresponding defined edge flow $F(s \to s') = \sum_{s \to s' \in \tau} F(\tau)$ and state flow $F(s) = \sum_{s \in \tau} F(\tau)$. The forward transition probabilities $P_F$ for each step of a trajectory can then be defined as Bengio et al. (2021b)

$$P_F\left(s \mid s'\right) = \frac{F\left(s \to s'\right)}{F(s)}.$$

GFlowNets aims to train a neural network to approximate the trajectory flow function with the output proportional to the reward function based on the flow matching condition Bengio et al. (2021b):

$$\sum_{s' \in \text{Parent}(s)} F\left(s' \to s\right) = \sum_{s'' \in \text{Child}(s)} F\left(s \to s''\right),$$

where $\text{Parent}(s)$ and $\text{Child}(s)$ denote the parent set and child set of state $s$, respectively. In this way, for any consistent flow $F$ with the terminating flow as the reward, i.e., $F\left(s \to s_f\right) = R(s)$ with $s_f$ being the final state and $s$ being the terminating state (can be transferred directly to the final state), a policy $\pi$ defined by the forward transition probability satisfies $\pi\left(s' \mid s\right) = P_F\left(s' \mid s\right) \propto R(x)$.

### 2.2 PROBLEM FORMULATION

A *multi-agent directed graph* is defined as a tuple $(\mathcal{S}, \mathcal{A})$ with $\mathcal{S}$ being a set of state and $\mathcal{A} = \mathcal{A}^1 \times \cdots \times \mathcal{A}^k$ denoting the set of joint edges (also called actions or transitions), which consists of all possible combinations of the actions available to each agent. A *trajectory* in such a graph is defined as a sequence $(s_1, ..., s_n)$ of elements of $\mathcal{S}$. A corresponding *multi-agent directed acyclic graph* (MADAG) is defined as a multi-agent directed graph with unequal pairs of states in the trajectory. Given an initial state $s_0$ and final state $s_f$, we name a trajectory $\tau = (s_0, ..., s_f) \in \mathcal{T}$ starting from $s_0$ and ending in $s_f$ as the complete trajectory, where $\mathcal{T}$ denotes the set of complete trajectories.

We consider a partially observable scenario, where the state $s \in \mathcal{S}$ is shared by all agents, but it is not necessarily fully observed. Hence, each agent $i \in \mathcal{I}$ selects an action $a^i \in \mathcal{A}^i$ based only on local observations $o^i$ made of a shared state $s \in \mathcal{S}$. In this way, we define the *individual edge/action flow* $F(o_t^i, a_t^i) = F(o_t^i \to o_{t+1}^i)$ as the flow through an edge $o_t^i \to o_{t+1}^i$, and the joint edge/action flow is defined by $F(s_t, \boldsymbol{a}_t) = F(s_t \to s_{t+1})$ with $\boldsymbol{a}_t = [a_t^1, ..., a_t^k]^T$. The state flow $F(s) : \mathcal{S} \mapsto \mathbb{R}$ is defined as $F(s) = \sum_{\tau \in \mathcal{T}} 1_{s \in \tau} F(\tau)$. Based on the flow matching condition Bengio et al. (2021b),

we have the state flow equal to the inflows or outflows, i.e.,

$$F(s) = \sum_{s', \boldsymbol{a}':T(s',\boldsymbol{a}')=s} F(s', \boldsymbol{a}') = \sum_{s' \in \text{Parent}(s)} F(s' \to s) \tag{1}$$

$$F(s) = \sum_{\boldsymbol{a} \in \mathcal{A}} F(s, \boldsymbol{a}) = \sum_{s'' \in \text{Child}(s)} F(s \to s''), \tag{2}$$

where $T(s', \boldsymbol{a}') = s$ denotes an action $\boldsymbol{a}'$ that can transfer state $s'$ to attain $s$. To this end, generative multi-flow networks (GMFlowNets) are defined as learning machines that can approximate trajectory flow functions in MADAG, with outputs proportional to the predefined reward function, trained based on flow matching conditions in equation 1 and equation 2.

# 3 GMFlowNets: Algorithms

## 3.1 Centralized Flow Network

Given such a MADAG, to train a GMFlowNet, a straightforward approach is to use a centralized training approach to estimate joint-flows, named Centralized Flow Network (CFN) algorithm, where multiple flows are trained together based on the flow matching conditions. In particular, for any state $s$ in the trajectory, we require that the inflows equal the outflows. In addition, the boundary condition is given by the flow passing through the terminating state $s$ based on the reward $R(s)$. Assuming we have a sparse reward setting, i.e., the internal states satisfy $R(s) = 0$ while the final state satisfies $\mathcal{A} = \emptyset$, then we have the flow consistency equation:

$$\sum_{s, \boldsymbol{a}:T(s,\boldsymbol{a})=s'} F(s, \boldsymbol{a}) = R(s') + \sum_{\boldsymbol{a}' \in \mathcal{A}(s')} F(s', \boldsymbol{a}'). \tag{3}$$

**Lemma 1** *Define a joint policy $\pi$ that generates trajectories starting in state $s_0$ by sampling actions $\boldsymbol{a} \in \mathcal{A}(s)$ according to*

$$\pi(\boldsymbol{a}|s) = \frac{F(s, \boldsymbol{a})}{F(s)}, \tag{4}$$

*where $F(s, \boldsymbol{a}) > 0$ is the flow through allowed edge $(s, \boldsymbol{a})$, which satisfies the flow consistency equation in equation 3. Let $\pi(s)$ be the probability of visiting state $s$ when starting at $s_0$ and following $\pi$. Then we have (a) $\pi(s) = \frac{F(s)}{F(s_0)}$; (b) $F(s_0) = \sum_{s_f} R(s_f)$; (c) $\pi(s_f) = \frac{R(s_f)}{\sum_{s'_f} R(s'_f)}$.*

**Proof:** The proof is trivial by following the proof of Proposition 2 in Bengio et al. (2021a).

We have Lemma 1, which shows that a joint flow function can produce $\pi(s_f) = R(s_f)/Z$ correctly when the flow consistency equation is satisfied. Then we can use a TD-like objective to optimize the joint flow function parameter $\theta$:

$$\mathcal{L}_\theta(\tau) = \sum_{s' \in \tau \neq s_0} \left( \sum_{s, \boldsymbol{a}:T(s,\boldsymbol{a})=s'} F_\theta(s, \boldsymbol{a}) - R(s') - \sum_{\boldsymbol{a}' \in \mathcal{A}(s')} F_\theta(s', \boldsymbol{a}') \right)^2. \tag{5}$$

Note that optimizing equation 5 is not straightforward. On the one hand, in each iteration, we need to estimate the flow in the order of $\mathcal{O}(|\mathcal{A}_i|^N)$[1], which leads to exponential complexity. The joint flow estimation method may get stuck in local optima and can hardly scale beyond dozens of agents. On the other hand, joint flow networks require all agents to sample jointly, which is impractical since in many applications the agents only have access to their own observations.

## 3.2 Independent Flow Network

To reduce the complexity and achieve the independent sampling of each agent, a simple way is to treat each agent as an independent agent, so that each agent can learn its flow function in the order of

---

[1]For simplicity, here we consider homogeneous agents, i.e., $\mathcal{A}_i = \mathcal{A}_j, \forall i, j \in N$. Moreover, heterogeneous agents also face the problem of combinatorial complexity.

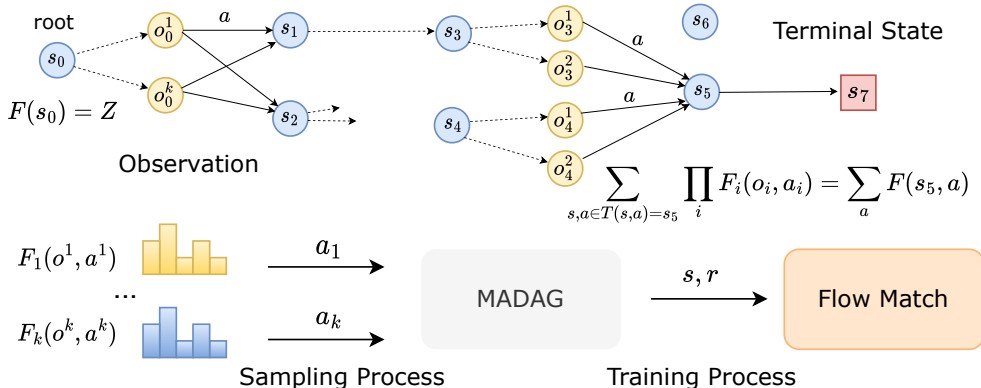

Figure 2: Framework of GMFlowNets. For each state, each agent obtains its own observation and computes its independent flow to sample actions. During training, the agent seeks the parent nodes for computing inflows and outflows, and performs policy optimization through flow matching.

$\mathcal{O}(|\mathcal{A}_i|)$. We call this approach the Independent Flow Network (IFN) algorithm, which reduces the exponential complexity to linear. However, due to the non-stationarity of the flow (see Definition 1), it is difficult for this algorithm to train a high-performance GMFlowNet.

**Definition 1 (Flow Non-Stationary)** *Define the independent policy $\pi_i$ as*

$$\pi_i(a_i|o_i) = \frac{F_i(o_i, a_i)}{F(o_i)}, \tag{6}$$

*where $a_i \in \mathcal{A}_i(o_i)$ and $F_i(o_i, a_i)$ is the independent flow of agent $i$. The flow consistent equation can be rewritten as*

$$\sum_{o_i, a_i : T(o_i, a_i, a_{-i}) = o'_i} F_i(o_i, a_i) = R(o_i, a_i) + \sum_{a'_i \in \mathcal{A}(o'_i)} F_i(o'_i, a'_i), \tag{7}$$

*where $-i$ represents other agents except agent $i$, and $R(o_i, a_i)$ represents the reward with respect to state $s$ and action $a_i$.*

Note that the transition function $T(o_i, a_i, a_{-i}) = o'_i$ in equation 7 is also related to the actions of other agents, which makes estimating parent nodes difficult. In addition, the reward of many multi-agent systems is the node reward $R(s)$, that is, we cannot accurately estimate the action reward $R(o_i, a_i)$ of each node. This transition uncertainty and spurious rewards can cause the flow non-stationary property. This makes it difficult to assign accurate rewards to each action, and thus, it is difficult to train independent flow network with a TD-like objective function.

As shown in Figure 1, compared to the centralized training method, it is almost difficult for the independent method to learn a better sampling policy. One way to improve the performance of independent flow networks is to design individual reward functions that are more directly related to the behavior of individual agents. However, this approach is difficult to implement in many environments because it is difficult to determine the direct relationship between individual performance and overall system performance. Even in the case of a single agent, only a small fraction of the shaped reward function aligns with the true objective.

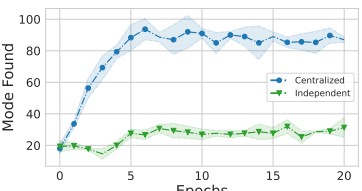

Figure 1: The performance of the centralized training and independent learning on Hyper-grid task.

### 3.3 FLOW CONSERVATION NETWORK

In this subsection, we propose the Flow Conservation Network (FCN) algorithm to reduce the complexity and simultaneously solve the flow non-stationary problem. FCN aims to learn the optimal

value decomposition from the final reward by back-propagating the gradients of the joint flow function $F$ through deep neural networks representing the individual flow function $F_i, \forall i \in N$. The specific motivation for FCN is to avoid flow non-stationary problems and reduce complexity. To start with, we have the following Definition 2, which shows the Individual Global Conservation (IGC) condition between joint and individual edge flows.

**Definition 2 (Individual Global Conservation)** *The joint edge flow is a product of individual edge flows, i.e.,*

$$F(s_t, \boldsymbol{a}_t) = \prod_i F_i(o_t^i, a_t^i).$$

Then, we propose the following flow decomposition theorem.

**Theorem 1** *Let the joint policy be the product of the individual policy $\{\pi_i\}_{i=1}^k$, where $\pi_i$ with respect to the individual flow function $F_i(o_i, a_i)$, i.e.,*

$$\pi_i(a_i|o_i) = \frac{F_i(o_i, a_i)}{F_i(o_i)}, \ \forall i = 1, \cdots, k. \tag{8}$$

*Assume that the individual flow $F_i(o_i, a_i)$ satisfies the condition in Definition 2. Define a flow function $\hat{F}$, if all agents generate trajectories using independent policies $\pi_i, \ i = 1, ..., k$ and the matching conditions*

$$\forall s' > s_0, \ \hat{F}(s') = \sum_{s \in \mathcal{P}(s')} \hat{F}(s \to s') \ and \ \forall s' < s_f, \ \hat{F}(s') = \sum_{s'' \in \mathcal{C}(s')} \hat{F}(s' \to s'') \tag{9}$$

*are satisfied. Then, we have:*
*1) $\pi(s_f) \propto R(s_f)$;*
*2) $\hat{F}$ uniquely defines a Markovian flow $F$ matching $\hat{F}$ such that*

$$F(\tau) = \frac{\prod_{t=1}^{n+1} \hat{F}(s_{t-1} \to s_t)}{\prod_{t=1}^{n} \hat{F}(s_t)}. \tag{10}$$

Theorem 1 states two facts. First, the joint state-action flow function $F(s, \boldsymbol{a})$ can be decomposed into the product form of multiple independent flows. Second, if any non-negative function satisfies the flow matching conditions, a unique flow is determined. On this basis, we can design algorithms for flow decomposition based on conservation properties. Each agent maintains a neural network to estimate the flow of its actions, then calculates the joint flow function through the flow conservation condition, and trains the model with the relevant reward function. In this case, each agent maintains a flow estimation network with the above architecture, which only estimates $(|\mathcal{A}_i|)$ flows. Compared with the centralized flow estimation network, we can reduce the complexity to $\mathcal{O}(N(|\mathcal{A}_i|))$. By combining $F_i(o_i, a_i)$, we can get an unbiased estimate of $F(s, \boldsymbol{a})$ to calculate a TD-like objective function. Next, we illustrate the overall training process.

During the individual sampling process, each agent samples trajectories using its own policy and composes a batch of data for joint training. During the joint training process, the system allows to call of the independent flow functions of each agent and uses the joint reward function to train the flow network. After training, each agent gets a trained independent flow network to meet the needs of independent sampling. In particular, for each sampled state, we first seek their parent nodes and corresponding observations and independent actions. Then, we compute the estimated joint flow $\hat{F}(s, \boldsymbol{a})$ by the flow consistency equation:

$$\hat{F}(s, \boldsymbol{a}) = \exp\left(\sum_{i=1}^k \log \hat{F}_i(o_i, a_i; \theta_i)\right), \tag{11}$$

where $\theta_i$ is the model parameter of the $i$-th agent, which can be trained based on equation 5 as:

$$\tilde{\mathcal{L}}(\tau; \theta) = \sum_{s' \in \tau \neq s_0} \left(\sum_{s, \boldsymbol{a}: T(s, \boldsymbol{a}) = s'} \hat{F}(s, \boldsymbol{a}) - R(s') - \sum_{\boldsymbol{a}' \in \mathcal{A}(s')} \hat{F}(s', \boldsymbol{a}')\right)^2. \tag{12}$$

Note that the above loss may encounter the problem that the magnitude of the flow on each node in the trajectory does not match, for example, the flow of the root node is large, while the flow of the leaf node is very small. To solve this problem, we here adopt the idea of log-scale loss introduced in Bengio et al. (2021a), and modify equation 12 as

$$\tilde{\mathcal{L}}(\tau, \epsilon; \theta) = \sum_{s' \in \tau \neq s_0} \left( \log \left[ \epsilon + \text{Inflows} \right] - \log \left[ \epsilon + \text{Outflows} \right] \right), \quad (13)$$

where

$$\text{Inflows} := \sum_{s, \boldsymbol{a}: T(s, \boldsymbol{a}) = s'} \exp \left[ \log \hat{F}(s, \boldsymbol{a}; \theta) \right] = \sum_{s, \boldsymbol{a}: T(s, \boldsymbol{a}) = s'} \exp \left[ \sum_{i=1}^{k} \log \hat{F}_i(o_i, a_i; \theta_i) \right]$$

$$\text{Outflows} := R(s') + \sum_{\boldsymbol{a}' \in \mathcal{A}(s')} \exp \left[ \log \hat{F}(s', \boldsymbol{a}'; \theta) \right] = R(s') + \sum_{\boldsymbol{a}'} \exp \left[ \sum_{i=1}^{k} \log \hat{F}_i(o_i', a_i'; \theta_i) \right],$$

and $\epsilon$ is a hyper-parameter that helps to trade-off large and small flows, which also avoids the numerical problem of taking the logarithm of tiny flows.

### 3.4 DISCUSSION: RELATIONSHIP WITH MARL

Interestingly, there are similar independent execution algorithms in the multi-agent reinforcement learning scheme. Therefore, in this subsection, we discuss the relationship between flow conservation networks and multi-agent RL. The value decomposition approach has been widely used in multi-agent RL based on IGM conditions, such as VDN and QMIX. For a given global state $s$ and joint action $\boldsymbol{a}$, the IGM condition asserts the consistency between joint and local greedy action selections in the joint action-value $Q_{\text{tot}}(s, \boldsymbol{a})$ and individual action values $[Q_i(o_i, a_i)]_{i=1}^k$:

$$\arg \max_{\boldsymbol{a} \in \mathcal{A}} Q_{\text{tot}}(s, \boldsymbol{a}) = \left( \arg \max_{a_1 \in \mathcal{A}_1} Q_1(o_1, a_1), \cdots, \arg \max_{a_k \in \mathcal{A}_k} Q_k(o_k, a_k) \right), \forall s \in \mathcal{S}. \quad (14)$$

**Assumption 1** *For any complete trajectory in an MADAG $\tau = (s_0, ..., s_f)$, we assume that $Q_{tot}^{\mu}(s_{f-1}, \boldsymbol{a}) = R(s_f) f(s_{f-1})$ with $f(s_n) = \prod_{t=0}^{n} \frac{1}{\mu(\boldsymbol{a}|s_t)}$.*

**Remark 1** *Although Assumption 1 is a strong assumption that does not always hold in practical environments. Here we only use this assumption for discussion analysis, which does not affect the performance of the proposed algorithms. A scenario where the assumption directly holds is that we sample actions according to a uniform distribution in a tree structure, i.e., $\mu(\boldsymbol{a}|s) = 1/|\mathcal{A}(s)|$. The uniform policy is also used as an assumption in Bengio et al. (2021a).*

**Lemma 2** *Suppose Assumption 1 holds and the environment has a tree structure, based on the IGC and IGM conditions we have:*
*1) $Q_{tot}^{\mu}(s, \boldsymbol{a}) = F(s, \boldsymbol{a}) f(s)$;*
*2) $(\arg \max_{a_i} Q_i(o_i, a_i))_{i=1}^k = (\arg \max_{a_i} F_i(o_i, a_i))_{i=1}^k$.*

Based on Assumption1, we have Lemma 2, which shows the connection between the IGC condition and the IGM condition. This action-value function equivalence property helps us better understand the multi-flow network algorithms, especially showing the rationality of the IGC condition.

## 4 RELATED WORKS

**Generative Flow Networks:** GFlowNets is an emerging generative model that could learn a policy to generate the objects with a probability proportional to a given reward function. Nowadays, GFlowNets has achieved promising performance in many fields, such as molecule generation Bengio et al. (2021a); Malkin et al. (2022); Jain et al. (2022), discrete probabilistic modeling Zhang et al. (2022) and structure learning Deleu et al. (2022). This network could sample the distribution of trajectories with high rewards and can be useful in tasks when the reward distribution is more diverse. This learning method is similar to reinforcement learning (RL) Sutton & Barto (2018), but

---

**Algorithm 1** Flow Conservation Network (FCN) Algorithm

---

**Input:** MADAG $\langle \mathcal{S}, \mathcal{A}, \mathcal{P}, \mathcal{R}, N \rangle$, Number of iteration $T$, Sample size $B$, Initial flow function $F_i^0, \forall i = 1, \cdots, k$, Parameters.
1: **for** iteration $t = 1, \cdots, T$ **do**
2:      \\ Individual sampling process
3:      Sample observations $\{(o_i^b, a_i'^{,b}, R^b)\}_{b=1}^B$ based on the individual flow function $F_i$ for all agents
4:      \\ Joint training process
5:      Seek all parent nodes $\{p^b\}$ of the global state $\{s^b\}_{b=1}^B$ and calculate the inflow $F(s^b, \boldsymbol{a}^b)$
6:      Calculate the outflow $Y^b = R^b(s) + \sum_{\boldsymbol{a} \in \mathcal{A}(s)} F(s, \boldsymbol{a})$ by the flow conservation condition
7:      Update the individual flow function: $\{\tilde{F}_i\} \leftarrow \arg\min_{\{F_i\}_{i=1}^k} \frac{1}{B}[Y^b - F(s^b, \boldsymbol{a}^b)]^2$
8: **end for**
9: Define the joint sampling policy as the product of the individual policies $\{\pi_i\}_{i=1}^k$ w.r.t. $\{F_i\}_{i=1}^k$
**Output:** flow function $\tilde{F}_T$ and individual sampling policy $\{\pi_i\}_{i=1}^k$

---

RL aims to maximize the expected reward and usually only generates the single action sequence with the highest reward. Conversely, the learned policies of GFlowNets can achieve that the sampled actions are proportional to the reward and are more suitable for exploration. This exploration ability makes GFNs promising as a new paradigm for policy optimization in the RL field, but there are many problems, such as solving multi-agent collaborative tasks.

**Cooperative Multi-agent Reinforcement Learning:** There are already many MARL algorithms to solve collaborative tasks, two extreme algorithms are independent learning Tan (1993) and centralized training. Independent training methods regard the influence of other agents as part of the environment, but the team reward function is usually difficult to measure the contribution of each agent, resulting in the agent facing a non-stationary environment Sunehag et al. (2017); Yang et al. (2020). On the contrary, centralized training treats the multi-agent problem as a single-agent counterpart. Unfortunately, this method exhibits combinatorial complexity and is difficult to scale beyond dozens of agents Yang et al. (2019). Therefore, the most popular paradigm is centralized training and decentralized execution (CTDE), including value-based Sunehag et al. (2017); Rashid et al. (2018); Son et al. (2019); Wang et al. (2020) and policy-based Lowe et al. (2017); Yu et al. (2021); Kuba et al. (2021) methods. The goal of value-based methods is to decompose the joint value function among agents for decentralized execution, which requires satisfying the condition that the local maximum of each agent's value function should be equal to the global maximum of the joint value function. VDN Sunehag et al. (2017) and QMIX Rashid et al. (2018) propose two classic and efficient factorization structures, additivity and monotonicity, respectively, despite the strict factorization method. QTRAN Son et al. (2019) and QPLEX Wang et al. (2020) introduce extra design for descomposition, such as factorization structure and advantage function. The policy-based methods extend the single-agent TRPO Schulman et al. (2015) and PPO Schulman et al. (2017) into the multi-agent setting, such as MAPPO Yu et al. (2021), which has shown the surprising effectiveness in cooperative, multi-agent games. The goal of these algorithms is to find the policy that maximizes the long-term reward, however, it is difficult for them to learn more diverse policies, which can generate more promising results.

## 5 EXPERIMENTS

We first verify the performance of CFN on a multi-agent hyper-grid domain where partition functions can be accurately computed. We then compare the performance of IFN and FCN with standard MCMC and some RL methods to show that their sampling distributions better match normalized rewards. All our code is implemented by the PyTorch Paszke et al. (2019) library. We reimplement the multi-agent RL algorithms and other baselines.

### 5.1 HYPER-GRID ENVIRONMENT

We consider a multi-agent MDP where states are the cells of a $N$-dimensional hypercubic grid of side length $H$. In this environment, all agents start from the initialization point $x = (0, 0, \cdots)$, and is only allowed to increase coordinate $i$ with action $a_i$. In addition, each agent has a stop action. When all agents choose the stop action or reach the maximum $H$ of the episode length, the entire

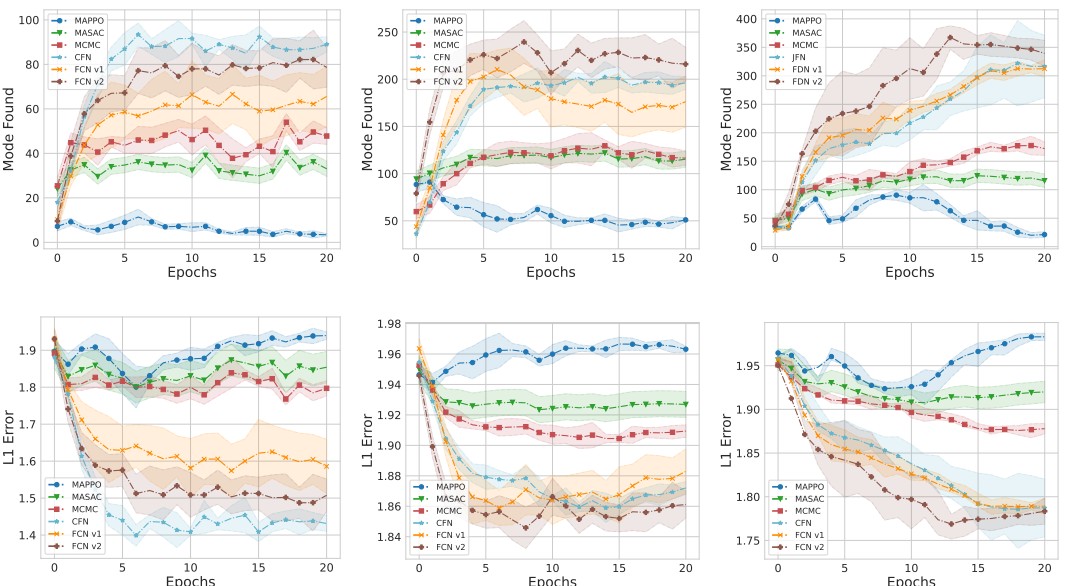

Figure 3: L1 error and Mode Found performance of different algorithms on various Hyper-grid environments. Top and bottom are respectively Mode Found (higher is better) and L1 Error (lower is better). **Left:** Hyper-Grid v1, **Middle:** Hyper-Grid v2, **Right:** Hyper-Grid v3.

system resets for the next round of sampling. The reward function is designed as

$$R(x) = R_0 + R_1 \prod_j \mathbb{I}\left(0.25 < |x_j/H - 0.5|\right) + R_2 \prod_j \mathbb{I}\left(0.3 < |x_j/H - 0.5| < 0.4\right), \quad (15)$$

where $x = [x_1, \cdots, x_k]$ includes all agent states, the reward term $0 < R_0 \ll R_1 < R_2$ leads a distribution of modes. By changing $R_0$ and setting it closer to 0, this environment becomes harder to solve, creating an unexplored region of state space due to the sparse reward setting. We conducted experiments in Hyper-grid environments with different numbers of agents and different dimensions, and we use different version numbers to differentiate these environments, the higher the number, the more the number of dimensions and proxies. Moreover, the specific details about the environments and experiments can be found in the appendix.

We compare CFN and FCN with a modified MCMC and RL methods. In the modified MCMC method Xie et al. (2021), we allow iterative reduction of coordinates on the basis of joint action space, and cancel the setting of stop actions to form a ergodic chain. As for RL methods, we consider the maximum entropy algorithm, i.e., multi-agent SAC Haarnoja et al. (2018), and a previous cooperative multi-agent algorithm, i.e., MAPPO, Yu et al. (2021). Note that the maximum entropy method uses the Softmax policy of the value function to make decision, so as to explore the state of other reward, which is related to our proposed algorithm. To measure the performance of these methods, we define the empirical L1 error as $\mathbb{E}[p(s_f) - \pi(s_f)]$ with $p(s_f) = R(s_f)/Z$ being the sample distribution computed by the true reward function. Moreover, we can consider the mode found theme to demonstrate the superiority of the algorithm.

| Environment | MAPPO | MASAC | MCMC | CFN | FCN v1 | FCN v2 |
|---|---|---|---|---|---|---|
| Hyper-Grid v1 | 2.0 | 1.84 | 1.78 | 2.0 | 2.0 | 2.0 |
| Hyper-Grid v2 | 1.90 | 1.76 | 1.70 | 1.85 | 1.85 | 1.82 |
| Hyper-Grid v3 | 1.84 | 1.66 | 1.62 | 1.82 | 1.82 | 1.78 |

Table 1: The best reward found of different methods.

Figure 3 illustrates the performance superiority of our proposed algorithm compared to other methods in the L1 error and mode found. For FCN, we consider two different decision-making methods,

the first is to sample actions independently, called FCN v1, and the other is to combine these policies for sampling, named FCN v2. We find that on small-scale environments shown in Figure 3-Left, CFN can achieve the best performance, because CFN can accurately estimate the flow of joint actions when the joint action space dimension is small. However, as the complexity of the joint action flow that needs to be estimated increases, we find that the performance of CFN degrades, but the independently executed method still achieves good estimation and maintains the speed of convergence, as shown in Figure 3-Middle. Note that RL-based methods do not achieve the expected performance, their performance curves first rise and then fall, because as training progresses, these methods tend to find the highest rewarding nodes rather than finding more patterns. In addition, as shown in Table 1, both the reinforcement learning method and our proposed method can achieve the highest reward, but the average reward of reinforcement learning is slightly better for all found modes. Our algorithms do not always have higher rewards than RL, which is reasonable since the goal of GMFlowNets is not to maximize rewards.

## 5.2 SMALL MOLECULES GENERATION

Similar to Jin et al. (2018); Bengio et al. (2021a); Xie et al. (2021), we consider the task of molecular generation to evaluate the performance of FCN. For any given molecular and chemical validity constraints, we can choose an atom to attach a block. The action space is to choose the location of the additional block and selecting the additional block. And the reward function is calculated by a pretrained model. We modify the environment to meets the multi-agent demand, where the task allows two agents to perform actions simultaneously depending on the state. Although this approach is not as refined as single-agent decision making, we only use it to verify the performance of FCN. Figure 4 shows that the number of molecules with the reward value greater than a threshold $\tau = 8$

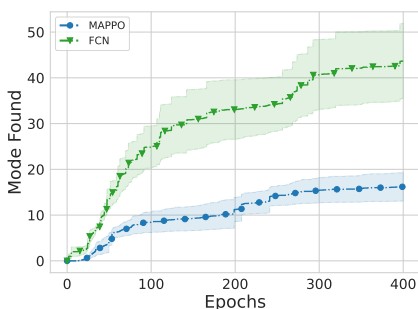

Figure 4: Performance of FCN and MAPPO on molecules generation task.

found by different algorithms, we can see that FCN can generate more molecules with high reward functions over three independent runs.

## 6 CONCLUSION

In this paper, we discuss the policy optimization problem when GFlowNets meet the multi-agent systems. Different from RL, the goal of GMFlowNets is to find diverse samples with probability proportional to the reward function. Since the joint flow is equivalent to the product of independent flow of each agent, we design a CTDE method to avoid the flow estimation complexity problem in fully centralized algorithm and the non-stationary environment in the independent learning process, simultaneously. Experimental results on Hyper-grid environments and small molecules generation task demonstrate the performance superiority of the proposed algorithms.

**Limitation and Future Work:** Unlike multi-agent RL algorithms that typically use RNNs as the value estimation network Hochreiter & Schmidhuber (1997); Rashid et al. (2018), RNNs are not suitable for our algorithms for flow estimation. The reason is that the need to compute the parent nodes of each historical state introduces additional overhead. Another limitation is that, like the original GFlowNets, GMFlowNets are constrained by DAGs and discrete environments, which makes GMFlowNets temporarily unavailable for multi-agent continuous control tasks. Therefore, our future work is to design multi-agent continuous algorithms to overcome the above problems.

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

# A    PROOF OF MAIN RESULTS

## A.1    PROOF OF THEOREM 1

**Theorem 1.** *Let the joint policy be the product of the individual policy $\{\pi_i\}_{i=1}^k$, where $\pi_i$ with respect to the individual flow function $F_i(o_i, a_i)$, i.e.,*

$$\pi_i(a_i|o_i) = \frac{F_i(o_i, a_i)}{F_i(o_i)}, \ \forall i = 1, \cdots, k. \tag{16}$$

*Assume that the individual flow $F_i(o_i, a_i)$ satisfies the condition in Definition 2. Define a flow function $\hat{F}$, if all agents generate trajectories using independent policies $\pi_i$, $i = 1, ..., k$ and the matching conditions*

$$\forall s' > s_0, \ \hat{F}(s') = \sum_{s \in \mathcal{P}(s')} \hat{F}(s \to s') \ and \ \forall s' < s_f, \ \hat{F}(s') = \sum_{s'' \in \mathcal{C}(s')} \hat{F}(s' \to s'') \tag{17}$$

*are satisfied. Then, we have:*
*1) $\pi(s_f) \propto R(s_f)$;*
*2) $\hat{F}$ uniquely defines a Markovian flow $F$ matching $\hat{F}$ such that*

$$F(\tau) = \frac{\prod_{t=1}^{n+1} \hat{F}(s_{t-1} \to s_t)}{\prod_{t=1}^{n} \hat{F}(s_t)}. \tag{18}$$

**Proof:** We first prove the part 1). Since

$$F(s_t, \boldsymbol{a}_t) = \prod_i F_i(o_t^i, a_t^i),$$

then we have the global state flow as

$$F(s_t) = \sum_{a_t \in \mathcal{A}} F(s_t, \boldsymbol{a}_t) = \sum_{\boldsymbol{a}_t \in \mathcal{A}} \prod_i F_i(o_t^i, a_t^i). \tag{19}$$

According to the flow definitions, the observation flow $F_i(o_t^i)$ and individual observation flows have the relationship:

$$F_i(o_t^i) = \sum_{a_t^i \in \mathcal{A}^i} F_i(o_t^i, a_t^i). \tag{20}$$

Hence, we have

$$\prod_{i=1}^{k} F_i(o_t^i) = \prod_{i=1}^{k} \left\{ \sum_{a_t^i \in \mathcal{A}^i} F_i(o_t^i, a_t^i) \right\} \tag{21}$$

$$= \sum_{a_t^1 \in \mathcal{A}^1} F_i(o_t^1, a_t^1) \cdots \sum_{a_t^k \in \mathcal{A}^k} F_i(o_t^k, a_t^k) \tag{22}$$

$$= \sum_{a_t^1, \cdots, a_t^k \in \mathcal{A}^1 \times \cdots \times \mathcal{A}^k} F_i(o_t^1, a_t^1) \cdots F_i(o_t^k, a_t^k) \tag{23}$$

$$= \sum_{a_t \in \mathcal{A}} \prod_{i=1}^{k} F_i(o_t^i, a_t^i), \tag{24}$$

yielding $F(s_t) = \prod_i F_i(o_t^i)$. Therefore, the joint policy

$$\begin{aligned}
\pi(\boldsymbol{a}|s) &= \frac{F(s_t, \boldsymbol{a}_t)}{F(s_t)} = \frac{\prod_i F_i(o_t^i, a_t^i)}{F(s_t)} \\
&= \frac{\prod_i F_i(o_t^i, a_t^i)}{\prod_i F_i(o_t^i)} = \prod_i \pi_i(a_i|o_i).
\end{aligned} \tag{25}$$

Equation 25 indicates that if the conditions in Definition 2 is satisfied, we can establish the consistency of joint and individual policies. Based on Lemma 1, we can conclude that the reward of the generated state satisfies $\pi(s_f) \propto R(s_f)$ using the individual policy $\pi_i(a_i|o_i)$ of each agent.

Next, we prove the part 2). We first prove the necessity part. According to Definition 2 and Bengio et al. (2021b) we have

$$F(s') = \prod_i F_i(o^{i,\prime}) = \prod_i \sum_{o^i \in \mathcal{P}(o^{i,\prime})} F_i(o^i \to o^{i,\prime}) = \sum_{\boldsymbol{o} \in \mathcal{P}(\boldsymbol{o'})} \prod_i F_i(o^i \to o^{i,\prime}),$$

$$F(s') = \prod_i F_i(o^{i,\prime}) = \prod_i \sum_{o^{i,\prime\prime} \in \mathcal{C}(o^{i,\prime})} F_i(o^{i,\prime} \to o^{i,\prime\prime}) = \sum_{\boldsymbol{o''} \in \mathcal{C}(\boldsymbol{o'})} \prod_i F_i(o^{i,\prime} \to o^{i,\prime\prime}).$$

Then we prove the sufficiency part. We first present Lemma 3, which shows that

$$\sum_{\tau \in \mathcal{T}_{0,s}} P_B(\tau) = \sum_{\tau \in \mathcal{T}_{0,s}} \prod_{s_t \to s_{t+1} \in \tau} P_B(s_t|s_{t+1}) = 1.$$

**Lemma 3 (Independent Transition Probability)** *Define the independent forward and backward transition respectively as*

$$P_F\left(o_{t+1}^i|o_t^i\right) := P_i\left(o_t^i \to o_{t+1}^i|o_t^i\right) = \frac{F_i\left(o_t^i \to o_{t+1}^i\right)}{F_i\left(o_t^i\right)}, \tag{26}$$

*and*

$$P_B\left(o_t^i|o_{t+1}^i\right) := P_i\left(o_{t+1}^i \to o_t^i|o_{t+1}^i\right) = \frac{F_i\left(o_{t+1}^i \to o_t^i\right)}{F_i\left(o_{t+1}^i\right)}. \tag{27}$$

*Then we have*

$$\begin{aligned}
\sum_{\tau \in \mathcal{T}_{s,f}} P_F(\tau) &= 1, \forall s \in \mathcal{S} \backslash \{s_f\}, \\
\sum_{\tau \in \mathcal{T}_{0,s}} P_B(\tau) &= 1, \forall s \in \mathcal{S} \backslash \{s_0\},
\end{aligned} \tag{28}$$

*where $\mathcal{T}_{s,f}$ is the set of trajectories starting in $s$ and ending in $s_f$ and $\mathcal{T}_{0,s}$ is the set of trajectories starting in $s_0$ and ending in $s$.*

Define $\hat{Z} = \hat{F}(s_0)$ as the partition function and $\hat{P}_F$ as the forward probability function. Then, according to Proposition 18 in Bengio et al. (2021b), we have there exists a unique Markovian flow $F$ with forward transition probability function $P_F = \hat{P}_F$ and partition function $Z$, and such that

$$F(\tau) = \hat{Z} \prod_{t=1}^{n+1} \hat{P}_F(s_t|s_{t-1}) = \frac{\prod_{t=1}^{n+1} \hat{F}(s_{t-1} \to s_t)}{\prod_{t=1}^n \hat{F}(s_t)}, \tag{29}$$

where $s_{n+1} = s_f$. Thus, we have for $s' \neq s_0$:

$$\begin{aligned}
F(s') &= \hat{Z} \sum_{\tau \in \mathcal{T}_{0,s'}} \prod_{(s_t \to s_{t+1}) \in \tau} \hat{P}_F(s_{t+1}|s_t) \\
&= \hat{Z} \frac{\hat{F}(s')}{\hat{F}(s_0)} \sum_{\tau \in \mathcal{T}_{0,s'}} \prod_{(s_t \to s_{t+1}) \in \tau} \hat{P}_B(s_t|s_{t+1}) = \hat{F}(s').
\end{aligned} \tag{30}$$

Combining equation 30 with $P_F = \hat{P}_F$, we have $\forall s \to s' \in \mathcal{A}, F(s \to s')$. Finally, for any Markovian flow $F'$ matching $\hat{F}$ on states and edges, we have $F'(\tau) = F(\tau)$ according to Proposition 16 in Bengio et al. (2021b), which shows the uniqueness property. Then we complete the proof.

### A.2 Proof of Lemma 2

**Lemma 2.** *Suppose Assumption 1 holds and the environment has a tree structure, based on the IGC and IGM conditions we have:*
*1) $Q_{tot}^{\mu}(s, \boldsymbol{a}) = F(s, \boldsymbol{a})f(s)$;*
*2) $(\arg\max_{a_i} Q_i(o_i, a_i))_{i=1}^{k} = (\arg\max_{a_i} F_i(o_i, a_i))_{i=1}^{k}$.*

**Proof:** The proof is an extension of that of Proposition 4 in Bengio et al. (2021a). For any $(s, \boldsymbol{a})$ satisfies $s_f = T(s, \boldsymbol{a})$, we have $Q_{tot}^{\mu}(s, \boldsymbol{a}) = R(s_f)f(s)$ and $F(s, \boldsymbol{a}) = R(s_f)$. Therefore, we have $Q_{tot}^{\mu}(s, \boldsymbol{a}) = F(s, \boldsymbol{a})f(s)$. Then, for each non-final node $s'$, based on the action-value function in terms of the action-value at the next step, we have by induction:

$$
\begin{aligned}
Q_{tot}^{\mu}(s, \boldsymbol{a}) &= \hat{R}(s') + \mu(\boldsymbol{a}|s') \sum_{\boldsymbol{a}' \in \mathcal{A}(s')} Q_{tot}^{\mu}(s', \boldsymbol{a}'; \hat{R}) \\
&\overset{(a)}{=} 0 + \mu(\boldsymbol{a}|s') \sum_{\boldsymbol{a}' \in \mathcal{A}(s')} F(s', \boldsymbol{a}'; R)f(s'),
\end{aligned}
\tag{31}
$$

where $\hat{R}(s')$ is the reward of $Q_{tot}^{\mu}(s, \boldsymbol{a})$ and $(a)$ is due to that $\hat{R}(s') = 0$ if $s'$ is not a final state. Since the environment has a tree structure, we have

$$
F(s, \boldsymbol{a}) = \sum_{\boldsymbol{a}' \in \mathcal{A}(s')} F(s', \boldsymbol{a}'),
\tag{32}
$$

which yields

$$
Q_{tot}^{\mu}(s, \boldsymbol{a}) = \mu(\boldsymbol{a}|s')F(s, \boldsymbol{a})f(s') = \mu(\boldsymbol{a}|s')F(s, \boldsymbol{a})f(s)\frac{1}{\mu(\boldsymbol{a}|s')} = F(s, \boldsymbol{a})f(s).
$$

According to the IGC condition we have $F(s_t, \boldsymbol{a}_t) = \prod_i F_i(o_t^i, a_t^i)$, yielding

$$
\begin{aligned}
\arg\max_{\boldsymbol{a}} Q_{tot}(s, \boldsymbol{a}) &\overset{(a)}{=} \arg\max_{\boldsymbol{a}} \log F(s, \boldsymbol{a})f(s) \\
&\overset{(b)}{=} \arg\max_{\boldsymbol{a}} \sum_{i=1}^{k} \log F_i(o_i, a_i) \\
&\overset{(c)}{=} \left( \arg\max_{a_1 \in \mathcal{A}_i} F_1(o_1, a_1), \cdots, \arg\max_{a_k \in \mathcal{A}_k} F_k(o_k, a_k) \right),
\end{aligned}
\tag{33}
$$

where $(a)$ is based on the fact $F$ and $f(s)$ are positive, $(b)$ is due to the IGC condition. Combining with the IGM condition

$$
\arg\max_{\boldsymbol{a} \in \mathcal{A}} Q_{tot}(s, \boldsymbol{a}) = \left( \arg\max_{a_1 \in \mathcal{A}_1} Q_1(o_1, a_1), \cdots, \arg\max_{a_k \in \mathcal{A}_k} Q_k(o_k, a_k) \right), \forall s \in \mathcal{S}.
\tag{34}
$$

we can conclude that

$$
\left( \arg\max_{a_i \in \mathcal{A}_i} F_i(o_i, a_i) \right)_{i=1}^{k} = \left( \arg\max_{a_1 \in \mathcal{A}_1} Q_i(o_i, a_i) \right)_{i=1}^{k}.
$$

Then we complete the proof.

### A.3 Proof of Lemma 3

**Lemma 3 [Independent Transition Probability].** *Define the independent forward and backward transition respectively as*

$$
P_F\left(o_{t+1}^i|o_t^i\right) := P_i\left(o_t^i \rightarrow o_{t+1}^i|o_t^i\right) = \frac{F_i\left(o_t^i \rightarrow o_{t+1}^i\right)}{F_i\left(o_t^i\right)},
\tag{35}
$$

*and*

$$
P_B\left(o_t^i|o_{t+1}^i\right) := P_i\left(o_{t+1}^i \rightarrow o_t^i|o_{t+1}^i\right) = \frac{F_i\left(o_{t+1}^i \rightarrow o_t^i\right)}{F_i\left(o_{t+1}^i\right)}.
\tag{36}
$$

*Then we have*

$$\sum_{\tau \in \mathcal{T}_{s,f}} P_F(\tau) = 1, \forall s \in \mathcal{S} \setminus \{s_f\},$$

$$\sum_{\tau \in \mathcal{T}_{0,s}} P_B(\tau) = 1, \forall s \in \mathcal{S} \setminus \{s_0\},$$

(37)

*where $\mathcal{T}_{s,f}$ is the set of trajectories starting in $s$ and ending in $s_f$ and $\mathcal{T}_{0,s}$ is the set of trajectories starting in $s_0$ and ending in $s$.*

**Proof:** When the maximum length of trajectories is not more than 1, we have

$$\sum_{\tau \in \mathcal{T}_{s,f}} P_F(\tau) = 1.$$

(38)

Then we have the following results by induction:

$$\sum_{\tau \in \mathcal{T}_{s,f}} P_F(\tau) = \sum_{s' \in \mathcal{C}(s)} \sum_{\tau \in \mathcal{T}_{s \to s', f}} P_F(\tau) = \sum_{o' \in \mathcal{C}(o)} P_F(o'|o) \sum_{\tau \in \mathcal{T}_{s', f}} P_F(\tau)$$

$$= \sum_{k} \sum_{o'_i \in \mathcal{C}(o_i)} P_F(o'_i|o_i) \sum_{\tau \in \mathcal{T}_{s', f}} P_F(\tau) = 1,$$

(39)

where $\mathcal{C}(\cdot)$ is the children set of the current state or observation and the last equation is based on the fact $\sum_{o'_i \in \mathcal{C}(o_i)} P_F(o'_i|o_i) = 1$. Since the proof process of $P_B$ is similar to that of $P_F$, it is omitted here.

## B  EXPERIMENTAL DETAILS

### B.1  HYPER-GRID ENVIRONMENT

Here we present the experimental details on the Hyper-Grid environments. Figure 5 shows the curve of the flow matching loss function with the number of training steps. The loss of our proposed algorithm gradually decreases, ensuring the stability of the learning process. For some RL algorithms based on the state-action value function estimation, the loss usually oscillates. This may be because RL-based methods use experience replay buffer and the transition data distribution is not stable enough. The method we propose uses an on-policy based optimization method, and the data distribution changes with the current sampling policy, hence the loss function is relatively stable. We set the same number of training steps for all algorithms for a fair comparison. Moreover, we list the key hyperparameters of the different algorithms in Tables 2 3 4 5.

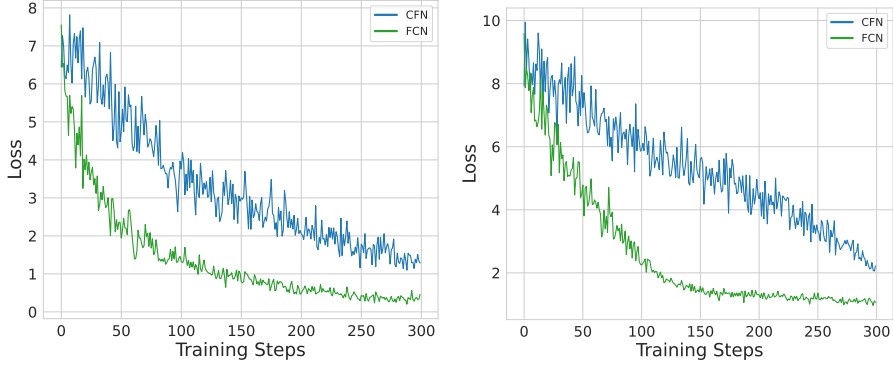

Figure 5: The flow matching loss of different algorithm.

We study the effect of different reward in Figure 6. In particular, we set $R_0 = \{10^{-1}, 10^{-2}, 10^{-4}\}$ for different task challenge. A smaller value of $R_0$ makes the reward function distribution more

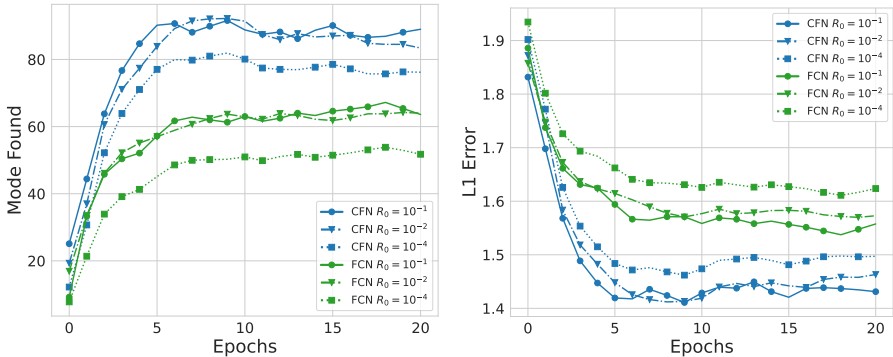

Figure 6: The effect of different reward $R_0$ on different algorithm according to L1 error and mode found.

sparse, which makes policy optimization more difficult Bengio et al. (2021a); Riedmiller et al. (2018); Trott et al. (2019). As shown in Figure 6, we found that our proposed method is robust with the cases $R_0 = 10^{-1}$ and $R_0 = 10^{-2}$. When the reward distribution becomes sparse, the performance of the proposed algorithm degrades slightly.

Table 2: Hyper-parameter of MAPPO under different environments

|  | Hyper-Grid-v1 | Hyper-Grid-v2 | Hyper-Grid-v3 |
| --- | --- | --- | --- |
| Train Steps | 20000 | 20000 | 20000 |
| Agent | 2 | 2 | 3 |
| Grid Dim | 2 | 3 | 3 |
| Grid Size | [8,8] | [8,8] | [8,8] |
| Actor Network Hidden Layers | [256,256] | [256,256] | [256,256] |
| Optimizer | Adam | Adam | Adam |
| Learning Rate | 0.0001 | 0.0001 | 0.0001 |
| Batchsize | 64 | 64 | 64 |
| Discount Factor | 0.99 | 0.99 | 0.99 |
| PPO Entropy | 1e-1 | 1e-1 | 1e-1 |

Table 3: Hyper-parameter of MASAC under different environments

|  | Hyper-Grid-v1 | Hyper-Grid-v2 | Hyper-Grid-v3 |
| --- | --- | --- | --- |
| Train Steps | 20000 | 20000 | 20000 |
| Grid Dim | 2 | 3 | 3 |
| Grid Size | [8,8] | [8,8] | [8,8] |
| Actor Network Hidden Layers | [256,256] | [256,256] | [256,256] |
| Critic Network Hidden Layers | [256,256] | [256,256] | [256,256] |
| Optimizer | Adam | Adam | Adam |
| Learning Rate | 0.0001 | 0.0001 | 0.0001 |
| Batchsize | 64 | 64 | 64 |
| Discount Factor | 0.99 | 0.99 | 0.99 |
| SAC Alpha | 0.98 | 0.98 | 0.98 |
| Target Network Update | 0.001 | 0.001 | 0.001 |

Table 4: Hyper-parameter of FCN under different environments

|  | Hyper-Grid-v1 | Hyper-Grid-v2 | Hyper-Grid-v3 |
|---|---|---|---|
| Train Steps | 20000 | 20000 | 20000 |
| $R_2$ | 2 | 2 | 2 |
| $R_1$ | 0.5 | 0.5 | 0.5 |
| Grid Dim | 2 | 3 | 3 |
| Grid Size | [8,8] | [8,8] | [8,8] |
| Trajectories per steps | 16 | 16 | 16 |
| Flow Network Hidden Layers | [256,256] | [256,256] | [256,256] |
| Optimizer | Adam | Adam | Adam |
| Learning Rate | 0.0001 | 0.0001 | 0.0001 |
| $\epsilon$ | 0.0005 | 0.0005 | 0.0005 |

Table 5: Hyper-parameter of CFN under different environments

|  | Hyper-Grid-v1 | Hyper-Grid-v2 | Hyper-Grid-v3 |
|---|---|---|---|
| Train Steps | 20000 | 20000 | 20000 |
| Trajectories per steps | 16 | 16 | 16 |
| $R_2$ | 2 | 2 | 2 |
| $R_1$ | 0.5 | 0.5 | 0.5 |
| Grid Dim | 2 | 3 | 3 |
| Grid Size | [8,8] | [8,8] | [8,8] |
| Flow Network Hidden Layers | [256,256] | [256,256] | [256,256] |
| Optimizer | Adam | Adam | Adam |
| Learning Rate | 0.0001 | 0.0001 | 0.0001 |
| $\epsilon$ | 0.0005 | 0.0005 | 0.0005 |

