# OpenReview forum: "Generative Multi-Flow Networks: Centralized, Independent and Conservation"
_ICLR.cc/2023/Conference — Submitted to ICLR 2023_

### Official Review · Reviewer_4dkX · 2022-10-19

**Confidence:** 4
**Correctness:** 1
**Technical Novelty And Significance:** 2
**Empirical Novelty And Significance:** 2
**Recommendation:** 3

**Clarity, Quality, Novelty And Reproducibility:**

Clarity and quality are very much under the expected level for ICLR. Besides the points raised above:
* the term IGM is used without having been defined (and it seems to have a pretty formal meaning, talking about IGM conditions, so it would be really important to define it)
* the second to last paragraph of the intro is very unclear
* Lemma 1 is just a reformulation of existing results in earlier GFlowNet papers
* In Definition 1, it seems strange to say that the agent reward "R(o_i, a_i) represents the reward with respect to state s and action a_i". Probably mean o_i rather than s?
* The term "flow non-stationary property" is used multiple times without explanation. It seems important but I do not know what it is.
* In Assumption 1, a function f(s_n) is defined in terms of a product of 1/mu(a|s_t). First, mu is never defined. Second, on the LHS we have only s_n while on the RHS we have s_1...s_n, which is generally not implied by s_n. Something is wrong!
* page 8, the term proxies is used: not clear what it means
* page 8, L1 error is defined as E[p(s_f) - pi(s_f)]. Presumable some absolute values are missing.

Although there might be novelty, it is so unclear because of the above issues that it is difficult to assess if it is meaningful.



**Strength And Weaknesses:**

The question of factorizing the policies (and the flows) of a GFlowNet is interesting and could be relevant in several domains, especially if this makes it possible to obtain efficient algorithms.

Unfortunately, I found the paper very poorly written, difficult to understand (both in the text and in the math), and left me with a major concern about the actual scalability of the proposed approach. The concern arises is Eq. 9, which requires performing sums over the cross-product of all the actions of all the agents (both forward or backward). The authors give no indication that these sums can be computed tractably: it looks like they require computation scaling exponentially with the number of agents. Unless there is a fix (and I did not see one even discussed), this makes the proposal not scalable.

Another issue is that the statement in definition 2 is not really a definition but rather an assumption, and more importantly, *it is not clear at all that under this assumption there generally exists a flow function* (that satisfies the flow matching constraints). This would make the mathematical foundation for all the theoretical results somewhat shaky. On page 5, the authors present this statement as as a FACT, "Theorem 1 states two facts. First, the joint state-action flow function F(s,a) can be decomposed into the product...". But it is not shown in theorem 1, it is an *assumption*.

Eqn 13 is wrong (it should have a square at the end).

There is also an incorrect statement about Bengio et al 2021a in Remark 1: "The uniform policy is also used as an assumption in Bengio et al 2021a", page 6. There is no such assumption in that paper. The only mention of uniform distribution in that paper is its use for defining an exploratory policy (which samples the training trajectories of the GFlowNet): they take a mixture of the GFlowNet policy P_F and the uniform, to make sure that all actions are given a non-zero probability. But that is no assumption at all, it is a heuristic to choose the training trajectories. The paper also has a theorem showing that any full-support distribution would asymptotically be sufficient to guarantee convergence to a proper flow. This is concerning because it suggests a misunderstanding of the original GFlowNet paper. It is also concerning because this argument is used to justify a completely different assumptio used here (the one stated as definition 2 discussed above).


**Summary Of The Paper:**

Several variants of GFlowNets are proposed in the context of multi-agent systems with centralized training and decentralized execution. The challenge considered in this paper is that we want to resulting policy to factorize across the agents, so that each agent can independently sample its next action given only a local observation.

**Summary Of The Review:**

The paper suffers from too many problems in both clarity of the math and of the text to be acceptable for publication. There may also be a fundamental flaw in the formulation which would make it unscalable (scale exponentially in compute time wrt number of agents).

---

### Official Review · Reviewer_pmcj · 2022-10-24

**Confidence:** 5
**Correctness:** 2
**Technical Novelty And Significance:** 2
**Empirical Novelty And Significance:** 2
**Recommendation:** 3

**Clarity, Quality, Novelty And Reproducibility:**

The main concern for the paper is its novelty and quality. Please see my comments in the above section.

**Strength And Weaknesses:**

- Clarity: The paper is easy to follow, although some of the terms (e.g., the experimental section) and the discussion of GFlowNets literature could be made clearer (discussion of the detailed balance and trajectory balance loss).

- Originality: The paper extends the GFlowNets (based on flow matching) to the multi-agent setting. Authors propose three kinds of methods: centralized flow network (related to directly training joint DQN in the MARL literature that faces the challenge of the curse of dimensionality), independent flow network (related to independent Q-learning in the MARL literature that faces the challenge of non-stationarity), and flow conservation network (related to CTDE methods in MARL like QMIX that allows for centralized training and decentralized execution). Although this is the first paper that extends GFlowNets to the multi-agent setting, I am concerned about its novelty. It combines existing methodology in the MARL literature and flow matching-based GFlowNets without enough in-depth analysis.

- Quality and significance: Besides the novelty concern, I am also concerned about the methodology part and the experiment part.
    - It has been shown that trajectory balance is one of the most competitive losses of GFlowNets, and it is worth discussing the possibility and potential of the proposed method based of trajectory balance.
    - I think the experimental benchmark is not very appropriate to evaluate multi-agent GFN (considering the single-agent feature of the experimental tasks including hypergrid and molecule generation originally used in the GFlowNets papers). I think it is more reasonable to evaluate multi-agent GFlowNets in standard multi-agent benchmarks such as StarCraft II micromanagement benchmarks.
    - Table 1 should also include the standard variation to better investigate different methods.
    - The details of the hypergrid task are not clear. For example, the author discuss the difficulty of changing R_0 closer to 0. It should be mentioned clearly in the paper about the size of the grid and the value of R_0, etc.
    - The comparison in Section 5.2 is somewhat weak, and should at least compare FCN with the most competitive method in Section 5.1 like MCMC and MASAC besides MAPPO.


**Summary Of The Paper:**

Considering recent success of GFlowNets, the paper aim to extend the GFlowNet framework to the multi-agent setting, which focuses on using the flow matching constraint. The authors propose three variants of multi-agent GFlowNets, including centralized flow network, independent flow network, and flow conservation network. The authors conduct experiments based on the hypergrid and molecule generation task to validate the effectiveness of the proposed methods by comparing it with MAPPO, MASAC and MCMC.

**Summary Of The Review:**

The paper extends the GFlowNets (based on flow matching) to the multi-agent setting. Although the paper is easy to follow, I think it can be improved greatly in terms of originality and quality. Although this is the first paper that extends GFlowNets to the multi-agent setting, I am concerned about its novelty. It combines existing methodology in the MARL literature and flow matching-based GFlowNets without enough in-depth analysis. The experimental benchmark is not very appropriate to fully investigate the value of multi-agent GFlowNets.

---

### Official Review · Reviewer_45Kq · 2022-10-24

**Confidence:** 5
**Correctness:** 2
**Technical Novelty And Significance:** 2
**Empirical Novelty And Significance:** 1
**Recommendation:** 5

**Clarity, Quality, Novelty And Reproducibility:**

There are many passages that could be clarified (see previous section).

As I write in the Weaknesses section, the paper combines existing ideas. This is not inherently bad, but in my opinion sets the bar higher. In addition, and assuming the best, the results are expected and unsurprising.

**Strength And Weaknesses:**

### Strengths

- The GFlowNet framework is very recent and there is value in exploring how it is compatible with other aspects of ML
- The proposed parameterizations make sense

### Weaknesses

- There seems to be something off with the evaluation. It would be good for the authors to clarify what is going on and if possible provide additional empirical validations of their approach
- There is no evaluation of the proposed method on standard multi-agent problems
- There are many lacking details and comments that aren't quite correct
- The proposed approaches are not particularly novel ideas but rather combinations of past ideas. This is not necessarily a bad thing, but in some sense the results and ideas presented in this work are expected and unsurprising, which naturally sets the bar higher.


### Details

> However, existing works can only handle single flow model tasks and cannot directly generalize to multi-agent flow networks due to limitations such as flow estimation complexity and independent sampling
> Unfortunately, currently GFlowNets cannot support multi-agent systems.

Interesting way of putting it, considering prior GFN work had no such ambitions. Saying something like, "because of its exploration capabilities, we extend the GFlowNet framework to support multi-agent systems" would make more sense.

The acronym IGM is used but not defined (I suppose it's Individual Global Maximum?).

> a policy $\pi$ defined by the forward transition probability satisfies $\pi(s'|s)=P_F(s'|s) \propto R(x)$

This is incorrect. A valid flow induces $p(x) = p(x\equiv s \to s_f) \propto R(x)$. In other words, when following policy $P_F(a|s)=F(s,a)/F(s)$ the probability of sampling a _trajectory_ which ends in $x$, i.e. a transition such that $x\equiv s$ is a terminal state, is proportional to the reward of $x$.

> unequal pairs of states in the trajectory

this is a slightly confusing phrasing, it might be easier to understand "such that $s_i \neq s_j \forall \tau,i\neq j$, but this is already the definition of an acyclic graph; I'm not sure this sentence is even needed.

> with outputs proportional to the predefined reward function

This is also somewhat incorrect, and it's not clear what "output" refers to. In the general case, for a non-terminal state with multiple parents and multiple children, state flows and edge flows are not likely to be proportional to some reward. Even for a terminal state, if this state has multiple parents, then only the sum of the incoming flows is equal to $R(x)$, not any particular flow on an edge leading to $x$. Similarly for trajectories, it is the sum of flows of trajectories which is equal to the reward, i.e. $\sum_{\tau:x\in\tau} F(\tau)=R(x)$, not the "output" for any one given trajectory.

> $s_f$

In Bengio et al. 2021a, terminal states are denoted by $x$, in Bengio et al. 2021b, the "GFlowNet Foundations" paper, the authors switch to using $s_f$ (see Definition 3 of a Pointed DAG), but it denotes a _single_ final sink state to which every terminating state transitions to, i.e. $s$ is terminating if $s \to s_f$ exists. I'd recommend keeping notation compatible with prior work.



> Definition 1 (Flow Non-Stationary)

Should be "Non-Stationarity"? Also, Definition 1 seems to define independent flows and policies, and says nothing of non-stationarity. Am I missing something?

> This transition uncertainty and spurious rewards can cause the flow non-stationary property

I agree that the partially observable system now appears _stochastic_ to any one agent and intuitively causes non-stationarity during learning, but, this "flow non-stationary property" is never defined. What is non-stationary? The reward? The transition function? The flow itself?


It seems counterproductive to introduce _two methods_ which are designed to fail, why not go straight to the point? It is well known in the multiagent literature that these problems exist (e.g. centralized execution leading to an exponential number of actions). I don't think this exposition was necessary.

> Definition 2, $F(s,a) = \prod_i F_i(o^i_t,a^i_t)$

I wonder what the partial observability assumptions are here. What must hold so that $F(s,.)$ is computable from individual observations $o^i$ and the product of $F_i$? This is non-trivial and is left unaddressed. Consider the common multiagent scenario of a (grid) world where agents' observations are restricted to some radius $r$ around them--it may be possible there that $F(s,a)$ is uncomputable under this parameterization, but it computable and Markovian under a joint/centralized parameterization.

> The uniform policy is also used as an assumption in Bengio et al. (2021a).

It is my understanding that the uniform policy is used in Proposition 4 of Bengio et al. (2021a) to show the equivalence _in a special case_ between a flow function and an action value function, and to conjecture that this equivalence _may_ be more general.

I'm also not sure I understand the point of Lemma 2. It seems trivial that if $f({\bf x}) = \prod f_i(x_i)$ then $\mbox{argmax}_ {\bf{x}} f(x) = \prod \mbox{argmax}_ {x_i} f_i (x_i)$ when $f_i>0$. If one assumes that the flow $F(s,{\bf a})$ is decomposable as a product, then trivially its argmax is a product of argmaxes. It follows from Proposition 4 of Bengio et al. (2021a) that there is an equivalence between $F$ and $Q$ on a tree-structured MDP. The IGM equivalence simply follows from assuming Definition 2.

It seems the attempt here is to provide an equivalence between the proposed flows and some $Q$ function as in Bengio et al. (2021a), or am I missing something deeper?

> the learned policies of GFlowNets can achieve that the sampled actions are proportional to the reward

This is also technically incorrect, see above; $p(x) \propto R(x)$, not $p($actions$)$ nor $p(\tau)$ (in the general case).

> The reward function is designed as [hypercube reward]

It seems this reward is taken from Bengio et al. (2021a), which it would be good to mention. Moreover, I'm not sure I understand what is multi-agent about this reward. All the agents get an independent reward, judging from the (incomplete) provided code, all the agents are allowed to move in all dimensions (it seems). The only explanation I can think of is that when the authors write "where $x = [x_1, ... , x_k]$ includes all agent states", they mean that in the $R$ computation $k$ is $k=N*m$ where $N$ is the hypercube dimension and $m$ is the number of agents.

It's also unclear in the main text nor the appendix what $m$ is exactly; I see it specified in Table 2 for MAPPO but not for the other algorithms, is it the same?

> we define the empirical L1 error as

This is again following the grid example of Bengio et al. (2021a), there seems to be an $\mbox{abs}$ or $|\cdot|$ missing in the expectation.

> Figure 3, L1 row

It makes very little sense to measure the L1 error for RL methods, that is not something they optimize for and not something we'd ever expect them to minimize even by accident.

I just looked at Figure 2 of Bengio et al. (2021a) and their L1 metric goes from about $5\times 10^{-4}$ all the way down to $10^{-5}$, so it drops about one and a half orders of magnitude, whereas the reported L1 error here seems to plateau very close to the initial values. The magnitude of the measure is also puzzling, how can an average absolute difference of things that are less than 1 (because they are probabilities) be greater than 1?

I'll admit this is worrying. Is the proposed method capable of learning $p(x)\propto R(x)$? These grid experiments should be very quick; would it be possible to show a setting where the L1 error gets low? Would it be possible to visualize the learned distribution to see if it matches what is expected? (cf. Bengio et al. (2021a) Figures 10,12,13)

It's also not mentionned how long one epoch is, how many samples does the model see?

> Figure 3, modes found row

What are FDN and JFN? Typos?

> Figure 4

I can imagine that two agents editing the same molecule is harder than the original setup, but the orders of magnitude are again a bit worrying. The original GFlowNet paper seems to find about 1500 modes with $R > 8$ within having seen 1M molecules. It's again not mentioned what an epoch is so it's hard to compare but even being generous about what one epoch might be, the contrast is worrying.

Let's suppose that this is expected. It would be nice to have more MARL baselines, and it would be nice to test the proposed method on actual standard multiagent problems. It would also be nice to show that the trained models actually are learning a flow (see Figure 16 of Bengio et al. 2021a or Nica et al [1])

As is, the empirical evaluation of the method is quite limited.

> GMFlowNets are constrained by DAGs and discrete environments, which makes GMFlowNets temporarily unavailable for multi-agent continuous control tasks

There is a simple fix to this, proposed by Bengio et al. (2021b), which is to augment the state description with the current timestep. This naturally forms a DAG because it is impossible for the agent to go back in time, therefore making it impossible to go back to a previous state.

Note on formatting: it seems all the non-inline citations are missing paretheses. Are you using the `\citep{...}` command?


[1] Evaluating Generalization in GFlowNets for Molecule Design, Andrei Cristian Nica, Moksh Jain, Emmanuel Bengio, Cheng-Hao Liu, Maksym Korablyov, Michael M. Bronstein, Yoshua Bengio, MLDD 2022


**Summary Of The Paper:**

This paper proposes to adapt the GFlowNet framework to a multi-agent setting. It studies the typical MARL parameterizations, and in line with past work proposes a Centralized Training Decentralized Execution formulation of a GFlowNet parameterization where each agent produces an independent flow, and where the joint flow is their product.

The authors show that such a formulation is compatible with the GFlowNet framework, and produce some empirical validation of their approach on multi-agent versions of a grid world task and a molecular generation task.

**Summary Of The Review:**

It is nice to see more research on GFlowNets, and there is certainly value in the potential diversity that they could bring to something like MARL methods. Unfortunately, the empirical validation of the approach needs to be strengthened and the paper made more clear. I do not think that the claims made by the paper are correctly validated.

---

### Official Review · Reviewer_YqtU · 2022-10-25

**Confidence:** 3
**Correctness:** 2
**Technical Novelty And Significance:** 2
**Empirical Novelty And Significance:** 2
**Recommendation:** 3

**Clarity, Quality, Novelty And Reproducibility:**

The overall writing leaves room for improvement, both in terms of the prose and the technical description. I encourage the authors to proofread further.

**Strength And Weaknesses:**

The GFlowNet formalism does not specify how the flow function F should be parametrized. Indeed, in a multi-agent setting where agents are expected to act independently with limited information, it can be a useful inductive bias to factorize the flow function. However, it is not clear what the contributions of this paper are besides the empirical validation that this straightforward factorization works. Note that CFN is the GFlowNet formalism applied naively to the multi-agent setup where all agents are modeled jointly, while IFN is another naïve application of the formalism where agents are modeled completely independently.

A scalability issue the authors do not address is the sum over |A_i|^N actions when calculating the in- and out-flow, which is required for both CFN and FCN. A paper made public in Jan. 2022 (https://arxiv.org/abs/2201.13259) proposes a new training objective for GFlowNets which circumvents this large sum. If the authors are motivated by the scalability issue in the multi-agent setup, I encourage them to include the aforementioned paper in their investigation.

Last but not the least, I have several questions regarding the experiments. The central claim is that CFN is not scalable which necessitates factorizing the flow estimator. However, FCN is not clearly better than CFN even on the hard Hyper-Grid environments (v2, v3). (Note that the legend says FDN and JFN.) It is not clear what the authors mean by “[combining] these policies for sampling” for FCN v2 whereas FCN v1 “[samples] actions independently”. I could not find further elaboration elsewhere in the paper.

Furthermore, based on table 4 and 5, both FCN and CFN use the same model architecture for the flow estimator. However, CFN has a single estimator while FCN has one per agent. Could the authors clarify if these two approaches use the same number of trainable parameters? If FCN were using N times more parameters where N is the number of agents, this would further weaken the empirical result.


**Summary Of The Paper:**

In a multi-agent environment, the joint action space can be exponential in the number of agents. A naïve application of the GFlowNet framework requires learning a flow function over |A_i|^N actions, which can be hard to optimize. Another naïve approach which treats each agent independently also fails due to difficulties in estimating the reward w.r.t. to a single agent.

The authors propose a factorized flow function F(s_t, a_t)=\prod_i F_i(o_t^i, a_t^i) where i runs over the agents. This allows an individual F_i to model only |A_i| actions at the expense of having to sample actions independently on every step.

Empirical results demonstrate the superiority of CFN and FCN over MCMC and RL methods in terms of number of modes found and L1 error. On harder tasks, GFlowNet-based methods also outperform RL and MCMC in terms of the best reward found.

**Summary Of The Review:**

This paper applies GFlowNets to multi-agent learning scenarios, which can be of interest to many in the community. However, the paper can benefit from a clearer motivation, better experiments, and a more polished delivery. I encourage the authors to rework the manuscript and submit again.

---

### Decision · Program_Chairs · 2023-01-20

**Decision:**

Reject

**Justification For Why Not Higher Score:**

The paper received strong and quite consistent criticism from all reviewers and the authors have not responded.

**Justification For Why Not Lower Score:**

N/A

**Metareview: Summary, Strengths And Weaknesses:**

The paper received strong and quite consistent criticism from all reviewers and the authors have not responded.